# The Effect of H^+^ Fluence Irradiation on the Optical, Structural, and Morphological Properties of ZnO Thin Films

**DOI:** 10.3390/ma17246095

**Published:** 2024-12-13

**Authors:** Alejandra López-Suárez, Yaser D. Cruz-Delgado, Dwight R. Acosta, Juan López-Patiño, Beatriz E. Fuentes

**Affiliations:** 1Instituto de Física, Universidad Nacional Autónoma de México, A.P. 20-364, Ciudad de México 01000, Mexico; davidcruzdelg@ciencias.unam.mx (Y.D.C.-D.); dacosta@fisica.unam.mx (D.R.A.); 2Departamento de Física, Facultad de Ciencias, Universidad Nacional Autónoma de México, Ciudad de México 04510, Mexico; juanchoslopez@gmail.com (J.L.-P.); beatriz.fuentes@ciencias.unam.mx (B.E.F.)

**Keywords:** zinc oxide, chemical spray pyrolysis, ion irradiation, fluence

## Abstract

Polycrystalline zinc oxide (ZnO) thin films were deposited on soda-lime glass substrates using the chemical spray pyrolysis method at 450 °C. The samples were irradiated with 8 keV H^+^ ions at three different fluences using a Colutron ion gun. The effects of the irradiation on the structural, morphological, and optical properties were studied with different techniques, including Rutherford Backscattering Spectrometry (RBS), X-ray diffraction (XRD), Scanning Electron Microscopy (SEM), and Ultraviolet and Visible Spectroscopy (UV–Vis). The results show that ion irradiation enhances crystallinity, narrowing the optical band gap. The changes in transmittance are related to defect formation within the material, which acts as light absorption and re-emission centers. A shifting of the film’s preferred growth orientation to the c-axis and changing the grain morphology and size distribution was detected. We observed an increase in the lattice parameters observed after irradiation, suggesting an expansion of the crystalline structure due to ions incorporation and defects within the ZnO crystal lattice. The morphological study shows an increase in the average size of the large particles after irradiation. This change is attributed to the emergence of defects and nucleation centers during irradiation. The average size of small particles remained relatively constant after irradiation, suggesting that small particles are more stable and less susceptible to external influences, resulting in fewer changes due to irradiation.

## 1. Introduction

What distinguishes human beings from the rest of living organisms is their capacity to modify raw materials to manufacture technological devices for specific applications. Throughout history, Materials Science has been at the forefront of technological advances; by discovering new materials and modifying or optimizing the existing ones, we have achieved an astonishing increase in quality of life, scientific research, information, and technologies. In the digitized modern world, semiconductor materials play a crucial and urgent role in human activities, making their research a fundamental area of modern science and even a matter of national security for some countries [1].

Thin films are distinguished by their highly reduced structure and thickness compared with bulk materials. These films, thin layers of deposited materials on solid substrates, have a thickness ranging from nanometers to micrometers. This significant reduction leads to singular physical and chemical properties compared with bulk materials, sparking a growing interest in thin film research. Their potential applications in various areas, including electronics, optoelectronics, nanotechnology, and surface science, offer a promising future. The unique properties of thin films, such as their conductivity and transparency, are fascinating and pique the curiosity of researchers and scientists [2].

Thin films are produced through a method called “Deposition”. This process involves the deposition of atoms, ions, or molecules on a solid surface or substrate in a highly controlled manner, resulting in a well-defined and structured layer [3]. This thin layer, which can be made of one or multiple materials, offers many possibilities. Its composition and structure can be tuned, allowing for modifying its electrical, optical, mechanical, morphological, chemical, and structural properties [2]. The most widely used thin film production technologies are characterized by the nature of their involved processes, which are purely physical, chemical, or a combination of both [4]. Within chemical processes, we can find the Spray Pyrolysis technique.

The Ultrasonic Spray Pyrolysis technique [5] is based on the pyrolytic decomposition of a metallic compound dissolved in a solution when sprayed on a previously heated substrate. The nozzle’s geometry and the gas pressure determine the dew’s shape, drop size distribution, and dew’s rate. These factors determine the film’s growing kinetics and final quality. Other essential control parameters that affect the film’s quality are the nature and temperature of the substrate, solution composition, solution flow speed, deposition time, and distance between substrate and nozzle. This technique includes ultrasonic technology for spraying liquid precursors in fine drops that are transported to the substrate and undergo pyrolysis to create a thin film.

Zinc oxide (ZnO) is a chemical compound widely known and studied for its versatility and the broad range of devices in which it is applied. At room temperature and atmospheric pressure, this compound is a white powder naturally found in the Zincite mineral, which crystallizes in the wurtzite-type hexagonal crystal system [6]. Zinc oxide is a semiconductor material with peculiar properties, making it suitable for optoelectronic devices that operate in the UV region due to its bandgap (3.37 eV at 300 K) and its high exciton binding energy (60 meV) [7]. One of the first properties that drew attention was the simultaneous high transmittance within the visible region and the low resistivity after doping ZnO with group III elements [7].

Different techniques exist for growing ZnO thin films, but obtaining the desired material properties can be complex. Thermal annealing, a standard modification method, can sometimes result in a broad particle size distribution. To overcome this challenge, ion irradiation offers a promising solution. This method allows for precise control over the properties of materials, including ZnO, by manipulating the irradiation parameters.

Ion Irradiation, a physical phenomenon involving the interaction between charged particles (ions) and a solid material [8,9], is crucial in Materials Science, Plasma Physics, and other fields. Its ability to induce substantial changes in material properties is not just theoretical but has practical applications. At its core, ion irradiation transfers energy and momentum from incident ions to target materials. When a particle (ion or electron) collides with a material’s atoms, it transfers part of its energy, modifying the physical properties of the target.

This study involved the deposition of ZnO thin films on soda-lime glass using the spray pyrolysis technique at 450 °C. Afterward, the samples were implanted at room temperature with 8 keV H^+^ ions at fluences of 1 × 10^14^ H^+^/cm^2^, 1 × 10^15^ H^+^/cm^2^, and 2 × 10^15^ H^+^/cm^2^ using a Colutron ion gun. The resulting film variations in structural, morphological, and optical properties were investigated.

The main objective of this study, which, as far as we know, has not been attempted in this way and is a novel approach in terms of irradiation effects, is to modify the fluence of H^+^ ions at low energies to achieve two distinct effects on the samples, depending on the fluence applied. These effects are: (a) to investigate the changes in the material resulting from the interaction between the H^+^ ions and the electrons in the film. This interaction causes excitations that can produce a process similar to the thermal spikes, effectively functioning as thermal annealing at a local level and improving material crystallization, and (b) to explore the changes in the film’s morphology by using H^+^ ions as projectiles to induce damage when a higher fluence is employed.

Our study, which focuses on modifying ZnO thin films with H^+^ as a projectile, is a significant departure from the few studies conducted on ZnO, not in thin films, but in the nanorods form [10].

On the other hand, the literature provides little information on the irradiation of ZnO with other light ions. The few works that we found were the studies conducted by Khawal et al. [11], who irradiate ZnO nanoparticles using Li^3+^ as a projectile, and the studies by Singh et al. [12], who use Ar to irradiate ZnO thin films.

## 2. Materials and Methods

Zinc oxide thin films were deposited on a borosilicate glass (substrate) using the ultrasonic spray pyrolysis technique. Substrate dimensions were 25 × 75 mm. After the deposition process, the substrate was cut into small pieces to produce samples for different characterizations. The precursor solution to grow ZnO thin films was prepared by dissolving 10.97 g of zinc acetate salts in 250 mL of deionized H_2_O. The mixture was heated at 40 °C and agitated for 15 min. Afterward, 50 mL of acetic acid was added to stabilize the solution and help in the transparency and complete dissolution of the salts used. This solution was slowly shaken to dissolve the material. With the presence of this acid, uniform and dense films are obtained, and the generation of hydroxides is inhibited. Subsequently, 700 mL of methanol was added to obtain 1000 mL of solution. From the solution above, 70 mL were atomized on the substrate, which was heated at 450 °C. The distance between the substrate and the atomizer was fixed at 30 cm. The sample that was not irradiated was called N-I (No-irradiated) or pristine. Finally, ZnO thin films were irradiated with 8 keV H^+^ ions, using three different fluences: 1 × 10^14^ H^+^/cm^2^ (1E14), 1 × 10^15^ H^+^/cm^2^ (1E15) y 2 × 10^15^ H^+^/cm^2^ (2E15). Irradiations were made with a Colutron-type ion gun model at the Science Faculty, UNAM. Projectile fluence and energy were chosen to ensure that the H^+^ ions interacted fully with the ZnO film and deposited energy into the material.

The following analysis techniques studied the variations in the physical properties of these thin films.

To understand ZnO thin films’ elemental composition and thickness, we subjected them to a characterization using the Rutherford Backscattering Spectrometry (RBS) technique. This involved the 3 MV Pelletron accelerator (National Electrostatics Corporation, Middleton, WI, USA), using alpha particles with an energy of 2.5 MeV and a 1 mm beam diameter at the Physics Institute, UNAM. Backscattered alpha particles were collected using a surface barrier detector at 167° to the ion beam direction. The resulting signals were then amplified by an electronic system comprising a preamplifier and an amplifier before being sent to a multichannel analyzer and a computer. Here, the RBS spectrum of each sample was generated and analyzed using SIMNRA software (Version 6.06), ensuring the determination of the elemental composition and the width of the sample [13]. The thickness of the ZnO thin film was 116.18 ± 5.81 nm, as calculated in Section 3.1

The microstructure of ZnO thin films was obtained through X-ray Diffraction (XRD), using K_α_ radiation from Cu (λ = 1.5406 Å) in the Bruker D8 Advance diffractometer (Bruker, Billerica, MA, USA). This method allowed us to identify the crystalline phase and lattice parameters from the X-ray diffraction patterns.

Optical Absorption Spectroscopy was used through a UV–Vis Agilent HP 8453 optical spectrometer (Agilent Technologies, Santa Clara, CA, USA) to obtain the bandgap value of ZnO thin films. Measurements were made at room temperature within the interval 190–1100 nm.

The sample morphology and grain size were analyzed using a FEG JEOL JSM-7800 Scanning Electron Microscope (SEM) (JEOL, Tokyo, Japan). The Software ImageJ (Version 1.54k) was employed to measure the grain size.

## 3. Results and Discussion

### 3.1. RBS Results

The elemental concentration of the ZnO thin films was obtained using the RBS technique. Figure 1 presents the experimental RBS (black dots) and the simulated SIMNRA (red line) spectra of a ZnO thin film prepared at 450 °C. The peaks corresponding to the elements that compose the sample (Zn and O) and the substrate (Si and O) are visible in the spectra. The oxygen peak at a higher channel corresponds to the oxygen that composes the ZnO; meanwhile, the one at a minor channel forms the matrix (SiO_2_). During the SIMNRA analysis, we observed Zn traces up to 300 nm, indicating diffusion into the matrix during the deposition process due to the heating involved. According to the RBS analysis, the ZnO thin film in Figure 1 has an elemental composition of Zn (49.5%) and O (50.5%), indicating that the sample is nearly stoichiometric.

The RBS technique was also used to measure the thickness of the ZnO film (X_ZnO_). As ions pass through the sample, they lose energy depending on their interaction with the electrons or the nuclei that compose the material. Therefore, the thin film thickness can be calculated using the following Equation [14]:(1)$$X_{ZnO}=\frac{{∆E}_{Zn}}{{\left[\varepsilon \right]}_{Zn}^{ZnO}}\;$$
where $\Delta E_{Zn}$ represents the difference between the energy of backscattered ions on the material’s surface and the energy of the backscattered ions after penetrating the sample. This energy was calculated using the Full Width at Half Maximum (FWHM) of the Zn peak (refer to Figure 1) and the stopping cross-section factor ${\left[\varepsilon \right]}_{Zn}^{ZnO}$ [14].
(2)$${\left[\varepsilon \right]}_{Zn}^{ZnO}=k_{Zn}\varepsilon_{ZnO,E_{0}\;}+\frac{1}{cos\;\theta_{2}}{\;\varepsilon }_{ZnO,kE_{Zn}}$$

In Equation (2), *k_Zn_*, *ε_ZnO_*_,*E*0_, and *ε_ZnO_*_,*kEZn*_ represent the zinc’s kinematical factor, the stopping power of ZnO for the incident energy *E*_0_ and the stopping power for the energy *k_E_*_0_, respectively. Meanwhile, *θ*_2_ = 13° is the exit angle of the backscattered alpha particles.

The thickness of the ZnO film was calculated using Equation (1), which gives a value of 116.18 ± 5.81 nm.

The changes in the physical properties of ZnO thin films that we will see below are expected to be mainly due to the electronic excitations and ionizations, as observed in the next section in Figure 2a, where the ionization process occurs primarily in the film zone. However, it is crucial to remain attentive to the contributions due to nuclear-stopping power (Figure 2c) and not to overlook their significance.

### 3.2. SRIM Simulations

The Monte Carlo method stands out in studying particle transport and its interaction with matter. It is based on prior knowledge of the probability of specific processes in a physical system.

The main characteristic of the Monte Carlo method is its statistical approach, which is based on sampling and statistical inferences. The interaction between particles is a probabilistic process since we can only have statistical approximations of their possible interactions. Within these interactions, the Monte Carlo method creates a stochastic model based on probability density functions that allow us to accurately represent the interactions, energy distribution of the particles, and the trajectory they follow. Considering that the initial conditions of the particle are known (angle of incidence, energy, and type of incident particle), the dispersion processes between particles can be simulated, with their respective energy losses, their trajectories, and the effects produced when the projectile interacts with the electrons and nuclei of the targets.

The SRIM (Stopping and Range of Ions in Matter) code [15] is a powerful tool for simulating and analyzing the behavior of ions as they traverse different materials. This code, widely used in particle physics and materials science, provides a visual representation of the paths of the individual ions and collects data on their interactions with the material.

Upon completion of the simulation, the SRIM code provides the distribution curves showing the depth at which ions are implanted in the material. The graphs illustrate the ions’ interactions with the material’s electrons (ionization) and nuclei (vacancies, defects). The code’s ability to handle complex interactions, such as multiple scattering and nuclear reactions, ensures a comprehensive analysis of ion implantation, making it a valuable tool for understanding particle behavior in materials.

The concentration of the H atoms in the ZnO thin film, the distance the projectiles can travel through the ZnO before losing their energy and coming to a stop (ion range), the electronic energy loss (excitation and ionization events), and the nuclear energy loss (vacancies and defects) obtained when an 8 keV H^+^ ion irradiates a 116.18 ± 5.81 nm ZnO thin film deposited over a glass borosilicate matrix were simulated with the SRIM code and are shown in Figure 2.

Figure 2a illustrates the ion range and the distribution of H^+^ ions interacting with the material. The vertical red line, positioned almost in the middle of the distribution, separates the film (located on the left side of the figure) from the substrate (on the right side). As the projectiles traverse the material, they interact with the electrons and nucleus of the material, causing electronic and nuclear energy loss, respectively. This energy loss process is dynamic, with the ions gradually losing their energy.

When an ion collides with a material, it transfers some of its energy to the atoms within it. The interaction is inelastic if the ions collide with electrons; however, collisions with nuclei are elastic. These processes result in the ion losing energy, which increases the likelihood of further collisions with the material’s nuclei. The collisions continue until the ion comes to a stop within the material. When the ion passes through the material, it ionizes or excites the atoms along its path, leaving behind a trail of excited atoms. These excited atoms possess enough energy to vibrate with significant amplitude without moving from their positions in the lattice. This vibration transfers energy to neighboring atoms, causing them to become excited and initiating a cascade of excitation among adjacent atoms. As local excitation develops, the state of the lattice changes as though a small region has been heated to high temperatures. In this manner, excitation propagation can be understood as a heat conduction process. This process can create extremely high temperatures in the surrounding region, resulting in a momentarily molten track. Due to the small area of the track, cooling occurs on the order of 10^−11^ s, leading to the solidification of the material. This results in forming a narrow cylindrical region with defects called columnar defects, ion tracks, or thermal spikes [16]. Consequently, this process leads to the growth of grains in the material, akin to the effects of thermal annealing. Thermal spikes have been observed with swift heavy ion irradiation. However, the effects produced by the thermal spikes are also discussed in light ions, as in the works of Khawal et al. [11], who irradiate ZnO nanoparticles using Li^3+^, Jeet et al., which shows how H^+^ ion irradiation can modify the structure of carbon nanotubes [17], Singh et al. [12], who use Ar to irradiate ZnO thin films, or Kucheyev et al., who used He^2+^ to irradiate silica [18].

Figure 2b,c present SRIM simulations of ionization and collision events resulting from the interaction of 8 keV H^+^ ions with a ZnO thin film deposited over a glass borosilicate matrix. These simulations, which show that ionization dominates over collisions at the surface, are significant. Most ionization and vacancy events, or the production of defects, occur within the ZnO thin film, indicating that the ion irradiation is causing alterations to the film. The subsequent sections of this work will delve into the implications of these material changes due to ion irradiation.

### 3.3. Structural Properties

X-ray Diffraction (XRD) is a technique based on the interaction between matter and electromagnetic radiation. When X-rays interact with the crystalline structure of materials, they undergo diffraction; thus, diffracted rays present constructive or destructive interference. Constructive interference takes place when the effects of two in-phase waves sum. Bragg’s law governs this phenomenon and is shown in Equation (3) [19]:(3)$$n\lambda =2d_{\left(hkl\right)}sin\left(\theta_{\left(hkl\right)}\right)$$
where *λ*, *d*_(_*_hkl_*_)_, (*hkl*), and *θ* are the X-ray wavelength (λ_Kα_ = 1.5406 Å), the interplanar spacing, the Miller indices, and the X-ray angle of incidence, respectively.

The lattice parameters of a hexagonal structure can be obtained using Equation (4), which relates the interplanar spacing with the Miller indices [20]:(4)$$\frac{1}{d_{\left(hkl\right)}^{2}}=\frac{3}{4}\;\left(\frac{h^{2}+hk+k^{2}}{a^{2}}\right)+\frac{l^{2}}{c^{2}}$$

The magnitude of a peak in a diffractogram reflects the number of diffracted X-rays in a specific direction. A higher intensity suggests a higher degree of diffraction by the crystal, which is related to the structure and the amount of material contributing to the diffraction. A high-quality crystal with a highly ordered structure tends to produce intense peaks, underscoring the need for meticulous crystal preparation. On the other hand, crystals with defects, tensions, or amorphous structures can produce peaks with lower intensity, highlighting the importance of crystal integrity. The Full Width at Half Maximum is a parameter calculated from the intensity of the most pronounced peaks in XRD. The grain size, shape, and degree of crystallinity can influence such a parameter.

Figure 3 depicts the crystalline structure of the ZnO thin films. The X-ray diffraction (XRD) analysis revealed peaks at (100), (002), (101), (102), and (103). These patterns indicate that the ZnO films have a polycrystalline structure with a hexagonal wurtzite shape and a preferred growth orientation along the c-axis.

The diffractograms demonstrate a clear pattern: as the H^+^ ions fluence increases, the peak intensity (002) also rises, except in the case of the 2E15 sample. This increase in intensity, a measure of the amount of material diffracting X-rays in a specific direction, indicates a significant enhancement in sample crystallinity. It suggests that irradiation is crucial in creating favorable conditions for nucleation and crystal growth, a phenomenon also observed by Krishna et al. [21]. The (002) peak of the pristine sample is located at 2θ = 34.6, but after irradiation, the diffraction peaks shift to a lower angle of 2θ = 34.3, as is verified in the inset of Figure 3. This shift reveals an expansion of the lattice parameters due to H^+^ irradiation. Chan et al. [22] observed a similar behavior in ZnO crystals irradiated with H^+^ ions. However, Wu et al. [10] observed the opposite. This discrepancy in the results could be attributed to the crystal form of the ZnO used in each study. Other studies also observed a shift to lower peak (002) angles with irradiation [11].

The pristine sample, which exhibits a slight amorphization at small angles, undergoes a significant transformation as fluence increases and is induced by irradiation. As the beam passes, it triggers crystallinity and particle nucleation. Figure 2b illustrates the Monte Carlo simulation of 8 keV H^+^ ions colliding with the ZnO thin film used in this work. The process of ionization and excitation of the atoms as the projectile passes through the material is robust, creating a thermal spike-like in the surrounding region where the ions move. This thermal spike-like, as was explained in Section 3.2, acts as a thermal treatment, repairing the crystal’s defects produced during the growth of the film.

JCPDS data (File 36-1451) [23] show that the lattice constants for a bulk ZnO material are *a* = 3.24982 Å and *c* = 5.20661 Å, where *c*/*a* = 1.6021. The lattice constants of the ZnO thin films were calculated using Equation (4), and the results are displayed in Table 1.

The results from Table 1 show a consistent trend in the lattice parameters after irradiation. The *a* and *c* values show a slightly increased value after irradiation compared with the non-irradiated sample. This increase in the lattice parameters indicates an expansion of the crystalline structure due to incorporating H atoms into the hexagonal structure. Our findings align with Khawal et al.’s [11], providing further validation. These results also concord with the shift to lower angles of the peak (002) observed in Figure 3 and reported before.

Using the (002) peak diffraction and the Scherrer formula, the average size (*D*) of the crystallites in the ZnO films was calculated by Equation (5) [24]:(5)$$D=\frac{k\;\lambda }{\beta \;cos\theta }$$
where k is the shape factor (0.9), *λ* is the wavelength of the Cu Kα, *β* is the FWHM of the most intense peak of the XRD curves, and *θ* is the Bragg angle.

The results in Table 2 show that the crystallite size increases with irradiation, except in the sample irradiated at the higher fluence. When the film is grown using the pyrolysis technique, crystals are formed, having an average size of 33.82 nm. In materials science, it is well known that the annealing process favors the nucleation of the particles and reduces the defects produced during the synthesis process. In our case, the irradiation process acts as the annealing process due to the thermal spike-like that forms during the ion’s trajectory through the material due to the dense electronic excitation deposited in a short time and a nanometric space.

The thermal spike-like increases the temperature of the material where the projectile’s track passes through and induces nanometric transformations of the irradiated material. Hence, it nucleates the minor crystals and increases the crystallite size. This behavior is observed in the samples irradiated at the lowest fluences. The film irradiated at the highest fluence presents the opposite behavior: the crystallite size is reduced with irradiation. This can be explained by the fact that more defects are created as the fluence increases, so the beam fragments the crystallites, reducing their size. Khawal et al. [11] irradiated ZnO with Li^3+^ and found that the crystallite size decreases when fluence increases. They argue that this conduct might be due to defects created during the irradiation and the difference in mass values.

### 3.4. Optical Properties

The transmittance and band gap properties were studied using UV–visible spectroscopy from 190 to 1100 nm. The results reveal that in the UV region, the transmittance value sharply decreases due to the band absorption edge of the ZnO thin films [25], as is depicted in Figure 4. In the visible region, the transmittance of the films decreases from 88 to 66% (measured at 500 nm wavelength) as the irradiation fluence increases, indicating that the ion irradiation strongly influences the irradiated samples. This result suggests that ion irradiation alters the material’s capacity for transmitting light. We observed that transmittance in the visible spectra of the irradiated samples decreased when compared with the pristine material. However, as we thought it would occur, we did not observe a direct relation between transmittance and ion fluence. The reduction in transmittance could be related to the emergence of defects and tensions within the crystalline structure of ZnO due to hydrogen ion bombardment. These defects can generate energy levels within the bandgap, affecting the material’s capacity for absorbing or transmitting light [25]. The additional energy levels lead to electronic transitions, which absorb light in specific wavelengths. Another cause of transmittance reduction might be that as the grain size and crystallite size decrease, more particles can be detected in the films, which causes an increase in grain boundary density, producing an enhancement in optical scattering and a decrease in transmittance values, as has been observed by Zhu et al. [26].

Furthermore, as we will explain, the reduction in transmittance within these spectral regions can be related to morphology and particle size changes due to irradiation. In the case of the sample with the highest fluence, the interaction of ions with the ZnO films fractured the crystals. As the crystal size diminishes, as is observed in Table 2, the material absorption decreases. In the case of the lowest fluence, ion irradiation did not damage the material much, so the transmittance value is around 70%. On the other hand, the medium fluence had enough energy to nucleate the crystals, increasing their size and decreasing the transmittance in the visible region. These results concord with the structural results, precisely the crystal size and morphology changes observed in this study’s previous and following sections.

The energy band gap measures the gap between the conduction and valence bands. This quantity has an impact on the optical and electronic properties of materials. To study the change in the band gap in the samples as they were irradiated, the Tauc relation was used [27]:(6)αhv=A(hv−Eg)12
where *A* is a constant, *E_g_* is the energy band gap, *h* is Planck’s constant, *hv* is the incident photon’s energy, and *α* is the absorption coefficient, defined as:(7)$$\alpha =-\frac{ln(T)}{x}=\frac{1}{x}ln\left(\frac{100}{transmittance\;percentage}\right)$$
where *T* is the transmittance and *x* refers to the sample thickness, calculated in Section 3.1, using the RBS technique.

If (*αhν*)^2^ vs. *hν* is plotted and a linear regression restricted to the absorption edge is made, the optical band gap is obtained as the intersection between such a line and the *hv*-axis. Figure 5 shows the values of the energy band gap of the pristine and irradiated samples.

The findings presented in Figure 5 demonstrate that the energy band gap value decreases as the ion fluence increases. This reduction can be attributed to various factors, which will be explained. The presence of bound states created by H^+^ ion irradiation [28] introduces additional energy levels within the band gap. These bound states may alter the allowed electronic transitions within the material, decreasing the band gap value. The reduction in band gap can also be linked to the presence of oxygen vacancies and interstitial zinc atoms, which induce changes in the crystalline structure and narrow the energy gap, as observed by Shasha et al. [29] and Abdel-Galil et al. [25]. A decrease in the band gap can significantly affect the design of optoelectronic devices, such as photodetectors or solar cells, based on irradiated ZnO films. Modifying and controlling the band gap through ion irradiation provides a promising strategy for adjusting the optical properties of a material to suit specific applications and requirements [11].

### 3.5. Morphological Properties

Figure 6 presents SEM micrographs of ZnO films. These images reveal the films’ dense, regular, and compact surfaces with various particle sizes and shapes. Notably, the regularity and compactness of the surfaces were consistent across the samples, indicating a high level of uniformity. We observed that the samples were composed of small and large particles, and to conduct a more precise morphological study, we categorized them into two size ranges: small (<100 nm) and large (>100 nm) particles (grains).

The results from the Scanning Electron Microscopy (Figure 6a–d) show that ion irradiation leads to nucleation and grain growth in the ZnO films. Additionally, the irradiation changes the grain morphology, transitioning the structure from rounded to flake-shaped particles. The non-irradiated sample displays small, rounded grains. Figure 6b,c depict the samples irradiated at the lower fluences, where the particles change from rounded-like to flake-like due to ion irradiation. In the case of the sample irradiated with the highest fluence, the particles appear to revert to a non-irradiated state, transforming from flakes to rounded-like forms. These morphological changes result from inelastic collisions between the sample’s electrons and the H^+^ ions used during irradiation, as reported by Khawal et al. [11] and Kumar et al. [30].

The ImageJ software was used to measure the grain size of the ZnO films. Figure 7 shows the average grain size of small and large particles. The small particles, indicated by blue dots, increased slightly in size from 71 to 80 nm as the fluence increased during irradiation. However, their size was not significantly altered by the irradiation process. On the other hand, the large particles, shown as red circles, experienced significant growth after ion irradiation, especially the sample irradiated at 1 × 10^14^ H^+^/cm^2^. This growth pattern was not directly related to fluence, a behavior also observed in XRD studies, where the crystallite size followed the same pattern. The average particle size for the non-irradiated sample (N-I) was 192 nm, whereas the samples irradiated at 1 × 10^14^, 1 × 10^15^, and 2 × 10^15^ H^+^/cm^2^ showed average grain sizes of 324, 256, and 245 nm, respectively. This considerable increase in size resulted directly from the interaction between H^+^ ions and the film, which, due to the thermal spike-like, caused the nucleation of atoms that formed the film and increased the clustering of large particles. As ion fluence increased, the nucleation of the particles was veiled by nuclear-stopping power, which created defects in the material, causing the grains that form the material to fragment and decrease in size [12].

It is essential to note the growth difference between small and large particles. The small particles’ excellent stability and lower vulnerability to ion incorporation and defect formation can explain this behavior. With their more compact structure and fewer defects, small particles are less vulnerable to these processes, leading to a growth pattern different from large particles. Therefore, due to this intricate interplay, H^+^ ion irradiation has a limited effect on small particle growth.

## 4. Conclusions

A study was conducted on ZnO thin films deposited onto soda-lime glass substrates using the chemical spray pyrolysis technique at a substrate temperature of 450 °C. The films were then irradiated with low-energy H^+^ ions at different fluences. This study examined the effects of the irradiation on the structural, morphological, and optical properties, leading to the following conclusions. One of the most significant results is the band gap reduction for irradiated samples compared to non-irradiated ones. This change in the band gap, which indicates the emergence of bound states due to the effects of H^+^ ions in the ZnO crystal lattice, has direct and practical implications for optoelectronic applications. It underscores the importance of band gap control for the performance and efficiency of devices such as photodetectors, solar cells, and light emitters. The observed changes in transmittance are related to defect formation within the material, which acts as light absorption and re-emission centers. The presence of defects leads to changes in the optical properties and opens new possibilities for applications in optical sensors and sensing devices. Regarding structural properties, an increase in the lattice parameters was observed after irradiation, suggesting an expansion of the crystalline structure due to ion incorporation and defects within the ZnO crystal lattice. These defects can act as nucleation centers, thereby promoting the growth of large particles. The lattice parameter changes indicate an alteration in the crystal structure, which has significant implications for the electrical conductivity and mechanical properties of ZnO thin films. The morphological characterization revealed a significant increase in the average size of the large particles after irradiation. This change is attributed to the emergence of defects and nucleation centers during irradiation, leading to the clustering and growth of large particles. These findings are important for applications requiring precise particle size and distribution control, such as optoelectronic and electronic device coatings. Interestingly, the average size of small particles remained relatively constant after irradiation, suggesting that small particles are more stable and less susceptible to external influences, resulting in fewer changes due to irradiation. Additionally, the stopping power (electronic and nuclear) of H^+^ ions in the ZnO film is responsible for the growth of crystallites and grains in the samples. To sum up, this research provides an integral vision of how H^+^ ion irradiation affects the optical, structural, and morphological properties of ZnO thin films. The collected results highlight the importance of irradiation’s influence on material properties and its impact on future applications such as devices and advanced technologies. These findings provide a basis for future research in materials development and manufacturing more efficient devices for various technological applications.

## Figures and Tables

**Figure 1 materials-17-06095-f001:**
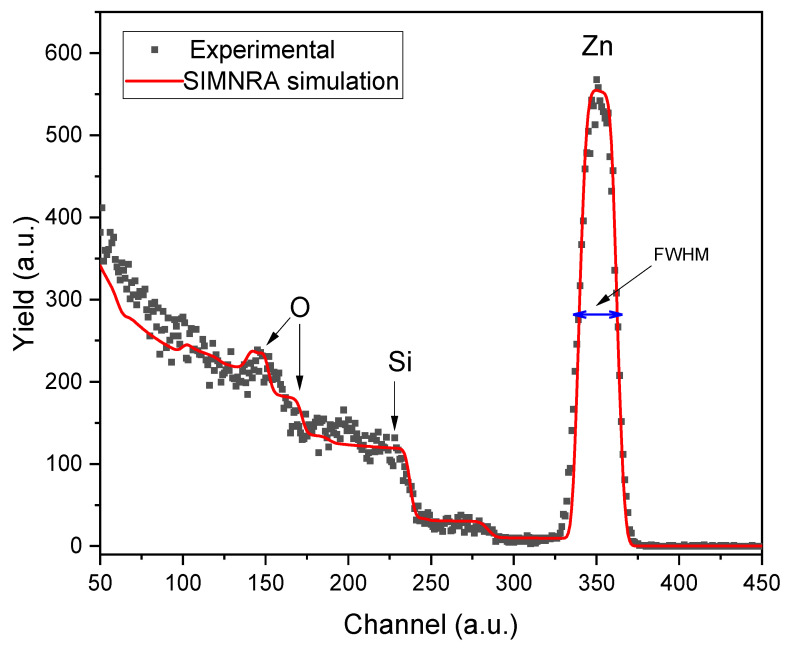
Experimental RBS (black dots) and simulated SIMNRA code (red line) spectra corresponding to the N-I ZnO thin film synthesized at 450 °C.

**Figure 2 materials-17-06095-f002:**
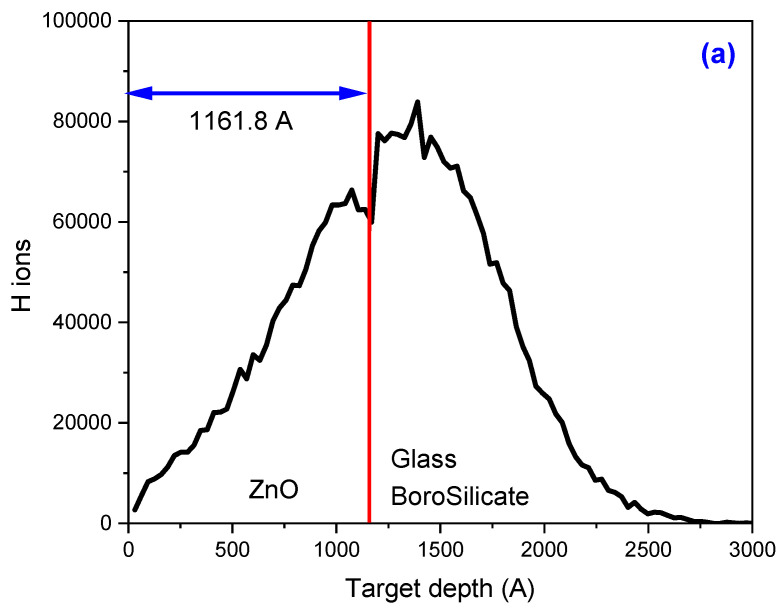
(**a**) Ion ranges, (**b**) ionization events, and (**c**) vacancies and defects produced during the irradiation of an 8 keV H^+^ ion into a 116.18 ± 5.81 nm ZnO thin film.

**Figure 3 materials-17-06095-f003:**
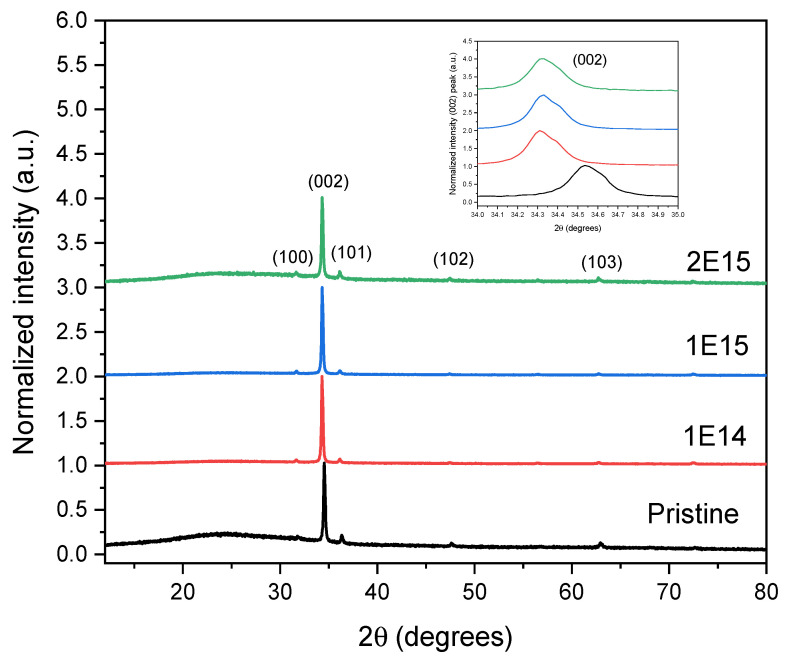
XRD Diffractograms of the irradiated and non-irradiated ZnO thin films. The inset illustrates the shift to lower angles in the irradiated samples than the pristine ones.

**Figure 4 materials-17-06095-f004:**
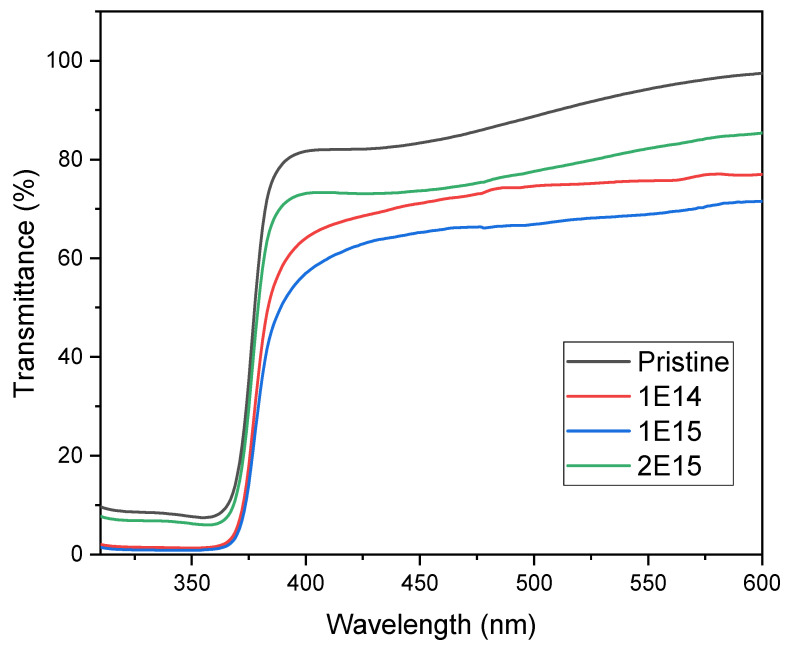
Optical transmittance spectra of the samples N-I (pristine), 1E14, 1E15, and 2E15.

**Figure 5 materials-17-06095-f005:**
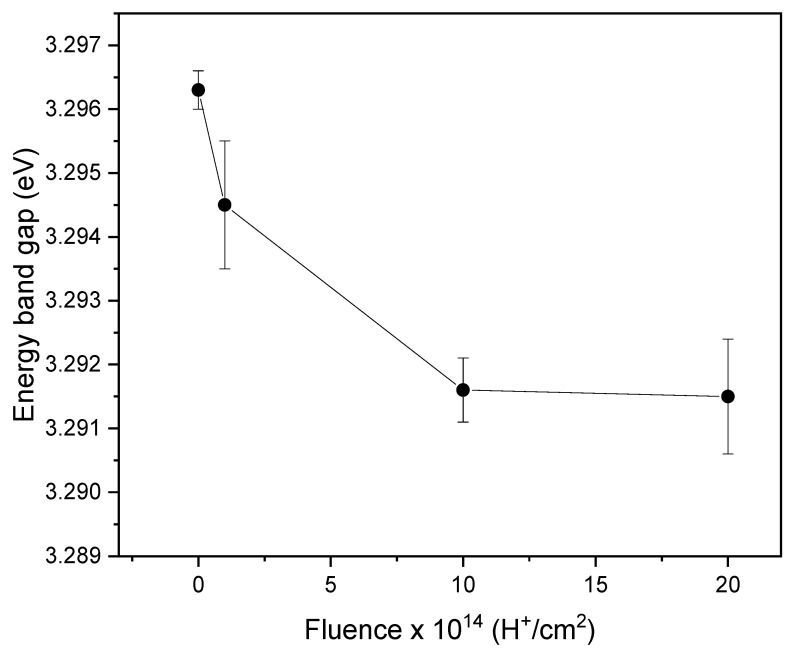
The energy band gap values of the pristine and irradiated ZnO samples.

**Figure 6 materials-17-06095-f006:**
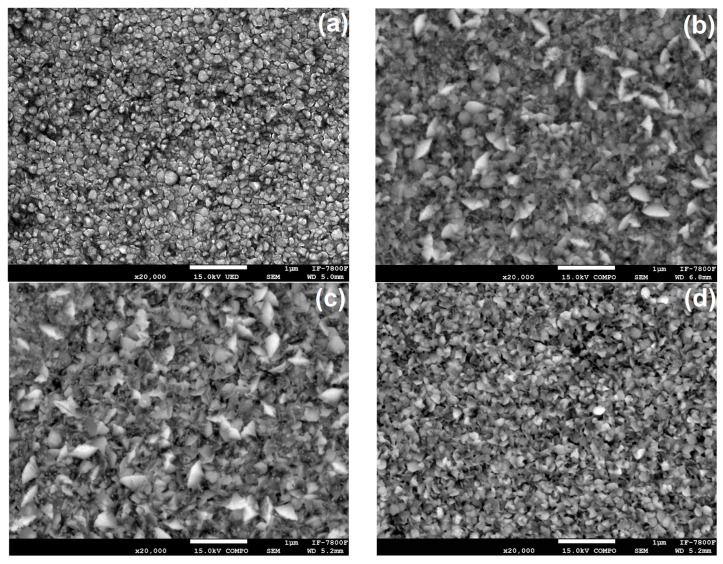
SEM micrographs of the (**a**) N-I, (**b**) 1E14, (**c**) 1E15, (**d**) 2E15 ZnO films.

**Figure 7 materials-17-06095-f007:**
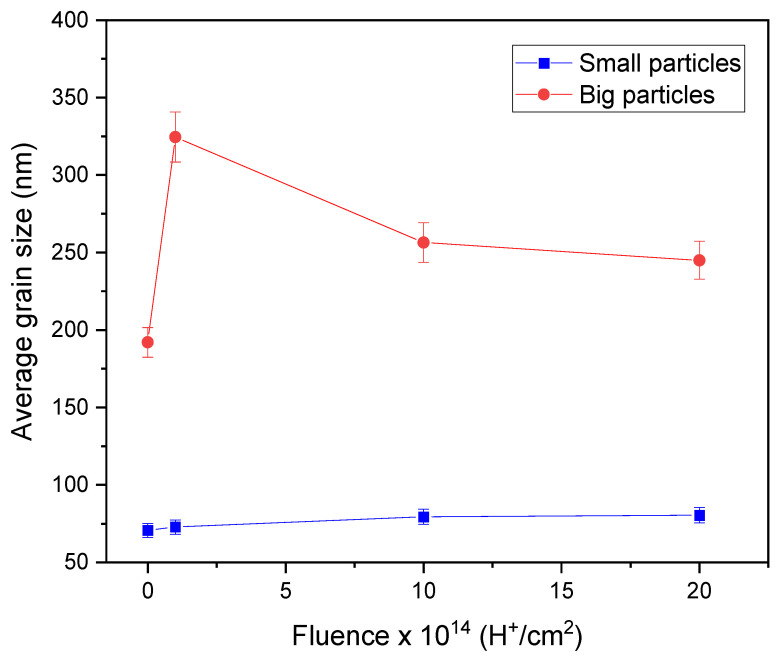
Dependence of average grain size on H^+^ fluence.

**Table 1 materials-17-06095-t001:** Lattice constants of pristine (N-I) and irradiated ZnO thin films.

Sample	H^+^ Fluence (H^+^/cm^2^)	a (Å)	c (Å)	c/a
Bulk ZnO JCPDS data [23]	0	3.24982	5.20661	1.6021
N-I	0	2.9963	5.1898	1.732
1E14	1 × 10^14^	3.0155	5.2230	1.732
1E15	1 × 10^15^	3.0137	5.2200	1.732
2E15	2 × 10^15^	3.0137	5.2200	1.732

**Table 2 materials-17-06095-t002:** The crystallite’s average size (D) in the non-irradiated and irradiated ZnO films.

Sample	Ion Irradiation Fluence (H^+^/cm^2^)	Crystallite’s Average Size (nm)
N-I	0	33.82
1E14	1 × 10^14^	40.54
1E15	1 × 10^15^	40.56
2E15	2 × 10^15^	36.86

## Data Availability

The raw data supporting the conclusions of this article will be made available by the authors on request due to bad experiences in the past when data of other work was available.

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
