# Peer review of "The Effect of H+ Fluence Irradiation on the Optical, Structural, and Morphological Properties of ZnO Thin Films"

_materials, 2024, doi:10.3390/ma17246095_

Round 1
Reviewer 1 Report
Comments and Suggestions for Authors
The authors synthesized ZnO thin films on soda-lime glass substrates by a chemical spray pyrolysis method and investigated the effects of H+ irradiation on the structural, optical and morphological properties of the ZnO films. I would recommend the publication of the paper after addressing the following issues.
1. An enlarged part near (002) peak as an inset in Figure 3 is needed to show the shift of the peak.
2. In figure 4, the value of y axis should be given for comparison.
3. Why can the morphology and particle size change the transmittance?
Author Response
We greatly appreciate your time and effort in reviewing this manuscript. In response, we have provided detailed answers below. In the re-submitted files, you will find all revisions and corrections highlighted in track changes. Thank you for your valuable feedback.
- An enlarged part near (002) peak as an inset in Figure 3 is needed to show the shift of the peak.
A: In the revised manuscript, Figure 3 shows an inset of an enlarged part near the (002) peak. This allows us to observe the shift of the (002) with ion irradiation. The manuscript text was changed to read, “The (002) peak of the pristine sample is located at 2θ= 34.6, but after irradiation, the diffraction peaks shift to a lower angle of 2θ= 34.3, as is verified in the inset of Figure 3.” The following text was added to Figure 3: “The inset illustrates the shift to lower angles in the irradiated samples compared to the pristine ones.”
- In figure 4, the value of y axis should be given for comparison.
A: Axis y in Figures 4(a) and 4(b) were included.
- Why can the morphology and particle size change the transmittance?
A: An explanation of this behavior has been added to section 3.4.
The variations in transmittance are due to several factors, including variations in sample crystallinity, crystallite size, and defects in the crystal lattice. In our case, we can combine the samples into two groups, depending on these factors. In the pristine and 2E15 samples, their crystallinity, grain size, and crystallite size are smaller than in samples 1E14 and 1E15, as shown in Figure 3, Figure 8, and Table 2, respectively. As the grain size and crystallite size decrease, more particles can be detected in the films, which causes an increase in grain boundary density, producing an enhancement in optical scattering and a decrease in transmittance values [B.L. Zhu, X.H. Sun, X.Z. Zhao, F.H. Su, G.H. Li, X.G. Wu, J. Wu, R. Wu, J. Liu. The effects of substrate temperature on the structure and properties of ZnO films prepared by pulsed laser deposition. Vacuum, 2008, 82, 495-500].
NOTE: Please see the attachment

Reviewer 2 Report
Comments and Suggestions for Authors
In revised paper The effect of H+ fluence irradiation on the optical, structural, and morphological properties of ZnO thin films authors firstly deposited ZnO layers by spray pyrolysis, irradiated it with hydrogen ions and investigated the influence of implantation on structural, optical and morphological properties. The presented topic is very interesting because precise and optimized ion implantation together with simple photolithography for hard mask creation could be used for optoelectronic device fabrication.
Below I listed my detailed remarks related with paper:
1. At the beginning, in the section Materials and method the projected thickness of ZnO is missing
2. What kind of lamp was used for transmittance measurements as a illumination source?
3. Line 155 - authors first refer the ZnO thickness then explain how the value was determined. This remark is in accordance with my 1st remarks, for me there is missing an initial (designed) thickness of ZnO estimated by process time
4. RBS measurement - Figure 2 - why authors measured RBS spectrum only for pristine sample? Why not for H+ implanted?
5. General remark - the measurement techniques description in my opinion is not necessary, all mentioned methods are widely used and authors can refer to paper or book or even their own previous work where same basis of the method is described
6. XRD results - Figure 3 - Is the intensity scale in log? The secondary peaks (100, 101, 102 and 103) are weakly visible. Moreover what is an origin of bump observed for pristine sample cantered at circa 25 degree?
7. XRD analysis - Table 1 - I would recommend to add row to the table with JCPDS data for ZnO
8. Lines 275 to 277 and Table 1 - why authors observed a divergence of lattice parameters of pristine sample when compared with JCPDS database?
9. Authors determined the crystallite size using a Scherrer equation. How the spray pyrolysis crystallite size corresponds to ZnO crystal size fabricated by different methods for instance ALD, PLD or magnetron sputtering?
10. For me it is a pity that authors did not investigate the basic electrical properties of pristine and implanted samples using 4 point probe or Hall/ECV measurements. However, maybe it could be a topic for next investigation
11. Similar remark with photoluminescence - it is a basic method suitable for defect investigation by measuring an emission spectrum of ZnO
12. Optical transmittance - Figure 5 - why below bandgap signal has different level, especially for pristine and 2E15 sample vs 1E14 and 1E15? This is really interesting due to fact the thickness of all sample is the same
13. Deterioration of optical properties (decrease of transmittance) could be an indicator of increase of defect concentration due to implantation. Mentioned PL could help to explain this phenomena
14. Structural characterization - SEM - did authors perform SEM observation of cross section? The observed of authors small particles could be top of buried big particles...
15. Last but not least - what was temperature of implantation process? If higher than room temperature, did authors consider implantation as thermal annealing?
Author Response
We greatly appreciate your time and effort in reviewing this manuscript. In response, we have provided detailed answers below. In the re-submitted files, you will find all revisions and corrections highlighted in track changes. Thank you for your valuable feedback.
- At the beginning, in the section Materials and method the projected thickness of ZnO is missing
A: The text: “The thickness of the ZnO thin film was 116.18±5.81 nm, as calculated in section 3.1.” was added to the manuscript.
- What kind of lamp was used for transmittance measurements as a illumination source?
A: An HP Agilent 8453 UV–VIS spectrophotometer was used to measure the transmittance. Its radiation source consists of a deuterium-discharge lamp for the ultraviolet (UV) wavelength range and a tungsten lamp for the visible and short-wave near-infrared (SWNIR) wavelength range.
- Line 155 - authors first refer the ZnO thickness then explain how the value was determined. This remark is in accordance with my 1st remarks, for me there is missing an initial (designed) thickness of ZnO estimated by process time
- Sections 3.1 and 3.2 were swapped to clarify the calculation of the film's thickness.
- RBS measurement - Figure 2 - why authors measured RBS spectrum only for pristine sample? Why not for H+ implanted?
A: Unfortunately, we could not measure the RBS spectrum for the implanted samples due to the Pelletron accelerator's unavailability after the irradiation. It's important to note that even if we had conducted RBS measurements, detecting any hydrogen signal would have been technically impossible due to its extremely small Rutherford cross-section.
- General remark - the measurement techniques description in my opinion is not necessary, all mentioned methods are widely used and authors can refer to paper or book or even their own previous work where same basis of the method is described.
A: We share your opinion, but we prefer to keep them since MATERIALS is a materials journal with such extensive topics that not all readers might be familiar with some of the techniques used in this study.
- XRD results - Figure 3 - Is the intensity scale in log? The secondary peaks (100, 101, 102 and 103) are weakly visible. Moreover what is an origin of bump observed for pristine sample cantered at circa 25 degree?
A: The intensity scale shown in Figure 3 is not logarithmic. The values have been normalized to the peak of the highest intensity, which in this case is the (002) peak, to facilitate a better comparison of the different spectra. Our focus in this work was primarily on the (002) peak, where crystal growth occurs. Therefore, we implemented normalization, even though this resulted in some loss of detail in the other peaks. The bump observed at 25 degrees indicates that the pristine crystal is slightly amorphous.
- XRD analysis - Table 1 - I would recommend to add row to the table with JCPDS data for ZnO
A: JCPDS data for bulk ZnO were added to Table 1, and the sentence “Based on JCPDS data (File 36-1451) [20], the lattice constants for a bulk ZnO material are a=3.24982 Å and c=5.20661 Å, where c/a = 1.6021.” was added to the text.
Table 1. Lattice constants of pritstine (N-I) and irradiated ZnO thin films.
Sample |
H+ fluence (H+/cm2) |
a (Å) |
c (Å) |
c/a |
Bulk ZnO JCPDS data [20] |
0 |
3.24982 |
5.20661 |
1.6021 |
N-I |
0 |
2.9963 |
5.1898 |
1.732 |
1E14 |
1×1014 |
3.0155 |
5.2230 |
1.732 |
1E15 |
1×1015 |
3.0137 |
5.2200 |
1.732 |
2E15 |
2×1015 |
3.0137 |
5.2200 |
1.732 |
- Lines 275 to 277 and Table 1 - why authors observed a divergence of lattice parameters of pristine sample when compared with JCPDS database?
A: The values ​​from the JCPDS database (File 36-1451) [20] correspond to a bulk ZnO crystal. Meanwhile, the lattice parameters of the pristine sample in this work correspond to a ZnO thin film.
- Authors determined the crystallite size using a Scherrer equation. How the spray pyrolysis crystallite size corresponds to ZnO crystal size fabricated by different methods for instance ALD, PLD or magnetron sputtering?
A: Crystallite size is not only affected by several parameters during the preparation of thin films using the spray pyrolysis technique. These parameters are the temperature and nature of the substrate, the spraying rate, the distance between the substrate and the spray nozzle, and the solvent used in the solution, which is directly related to the solution's molarity. However, the deposition temperature is the most critical parameter for changing the crystallite size in virtually all thin film deposition techniques.
Comparing the crystal size of a ZnO thin film using various deposition techniques is complicated due to the different parameters involved during sample growth; however, some examples are provided.
Deposition technique |
Crystallite size (nm) |
Reference |
This work (spray pyrolysis) |
33.82 |
|
The sol-gel spin coating method |
22.84 |
K.L. Foo, M. Kashif, U. Hashim, Wei-Wen Liu. Effect of different solvents on the structural and optical properties of zinc oxide thin films for optoelectronic applications. Ceramics International 2024, 40, 753-761. |
RF magnetron sputtering |
33 |
Ruitao Wen, Laisen Wang, Xuan Wang, Guang-Hui Yue, Yuanzhi Chen, Dong-Liang Peng. Influence of substrate temperature on mechanical, optical and electrical properties of ZnO:Al films. Journal of Alloys and Compounds 2010, 508, 370-374. |
Aerosol-assisted chemical vapor deposition (AACVD) |
72 |
Dominic B. Potter, Ivan P. Parkin, and Claire J. Carmalt. The effect of solvent on Al-doped ZnO thin films deposited via aerosol-assisted CVD. RSC Advances 2018, 8, 33164. |
Cathodic potentiostatic electrochemical deposition |
33.3 to 60.5, depending on temperature deposition. |
R.E. Marotti, P. Giorgi, G. Machado, E.A. Dalchiele. Crystallite size dependence of band gap energy for electrodeposited ZnO grown at different temperatures. Solar Energy Materials and Solar Cells 2006, 90, 2356-2361. |
- For me it is a pity that authors did not investigate the basic electrical properties of pristine and implanted samples using 4 point probe or Hall/ECV measurements. However, maybe it could be a topic for next investigation
A: We appreciate your feedback and will consider it for future research.
- Similar remark with photoluminescence - it is a basic method suitable for defect investigation by measuring an emission spectrum of ZnO.
A: We appreciate your feedback and will consider it for future research.
- Optical transmittance - Figure 5 - why below bandgap signal has different level, especially for pristine and 2E15 sample vs 1E14 and 1E15? This is really interesting due to fact the thickness of all sample is the same
A: Another explanation of this behavior has been added to section 3.4.
The variations in transmittance are due to several factors, including variations in sample crystallinity, crystallite size, and defects in the crystal lattice. In our case, we can combine the samples into two groups, depending on these factors. In the pristine and 2E15 samples, their crystallinity, grain size, and crystallite size are smaller than in samples 1E14 and 1E15, as shown in Figure 3, Figure 8, and Table 2, respectively. As the grain size and crystallite size decrease, more particles can be detected in the films, which causes an increase in grain boundary density, producing an enhancement in optical scattering and a decrease in transmittance values [B.L. Zhu, X.H. Sun, X.Z. Zhao, F.H. Su, G.H. Li, X.G. Wu, J. Wu, R. Wu, J. Liu. The effects of substrate temperature on the structure and properties of ZnO films prepared by pulsed laser deposition. Vacuum, 2008, 82, 495-500].
- Deterioration of optical properties (decrease of transmittance) could be an indicator of increase of defect concentration due to implantation. Mentioned PL could help to explain this phenomena
A: Unfortunately, we did not conduct PL measurements in this work. However, modifications such as defects and ionization in the sample properties were anticipated due to implantation, as suggested by implantation theory and the Monte Carlo simulations we performed. Structural and morphological results also indirectly show the defects produced during the implantation.
- Structural characterization - SEM - did authors perform SEM observation of cross section? The observed of authors small particles could be top of buried big particles...
A: We did not perform SEM observation of the cross-section. A cross-section study would have only allowed us to analyze a tiny part of the sample. This outline would not have been representative due to beam instability, which could sometimes occur during ion irradiation and is observed at the edge of the sample.
- Last but not least - what was temperature of implantation process? If higher than room temperature, did authors consider implantation as thermal annealing?
A: The implantation was carried out at room temperature; however, the thermal spike can considerably raise the temperature along the beam path. This increase in temperature may have effects similar to thermal annealing, as discussed in sections 3.2, 3.3, and 3.5.
Section 3.2's explanation of the thermal spike process was rewritten to be more comprehensible to the audience.
When an ion collides with a material, it transfers some of its energy to the atoms within it. The interaction is inelastic if the ions collide with electrons; however, collisions with nuclei are elastic. These processes result in the ion losing energy, which increases the likelihood of further collisions with the material's nuclei. The energy transferred from the ions to the nuclei is more significant than that transferred to the electrons. The collisions continue until the ion comes to a stop within the material. If the ion has a high velocity, it may pass through the material, ionizing or exciting the atoms along its path and leaving behind a trail of excited atoms. These excited atoms possess enough energy to vibrate with significant amplitude without moving from their positions in the lattice. This vibration transfers energy to neighboring atoms, causing them to become excited and initiating a cascade of excitation among adjacent atoms. As local excitation develops, the state of the lattice changes as though a small region has been heated to high temperatures. In this manner, excitation propagation can be understood as a heat conduction process. This process can create extremely high temperatures in the surrounding region, resulting in a momentarily molten track. Due to the small area of the thermal spike, cooling occurs on the order of 10×-11 s, leading to the solidification of the material. This results in forming a narrow cylindrical region with defects called columnar defects, ion tracks, or thermal spikes [D. Kauomi, A.T. Motta and R.C. Birtcher. A thermal spike model grain growth under irradiation. Journal of Applied Physics 2008, 104, 073525-13]. Consequently, this process leads to the growth of grains in the material, akin to the effects of thermal annealing.
NOTE: Please see the attachment

Round 2
Reviewer 2 Report
Comments and Suggestions for Authors
I am satisfied with corrections made by authors.